# Preparation and Performance Test of the Super-Hydrophobic Polyurethane Coating Based on Waste Cooking Oil

**Yingmo Cheng [1,2], Dejun Miao [1,2,*], Lingxiao Kong [1,2], Jiachen Jiang [1,2] and Zhenxing Guo [1,2]**

[1] School of Mining and Safety Engineering, Shandong University of Science and Technology, Qingdao 266590, China; chengyingmo1994@163.com (Y.C.); lingxiao_ko@163.com (L.K.); jjc17854252173@163.com (J.J.); gzx951005@163.com (Z.G.)

[2] National Key Laboratory Breeding Base for Mine Disaster Prevention and Control, Shandong University of Science and Technology, Qingdao 266590, China

\* Correspondence: miaodj@sdust.edu.cn

**Abstract:** In order to solve the problem of dust accumulation on the fin surface of a mine air cooler, a method of preparing super-hydrophobic polyurethane (SPU) coating based on waste cooking oil (WCO) was proposed. Firstly, the polyurethane prepolymer was synthesized with WCO as a raw material, and then the polyurethane prepolymer was modified with amino-terminated polydimethylsiloxane (ATP) to obtain SPU emulsion. The chemical structure and thermal stability of SPU were characterized by infrared spectrum and thermogravimetric analysis. A series of nanocomposites were prepared by combining modified silicon carbide (APT-SiC) particles and SPU emulsions. According to the parameters of pull-off strength, contact angle, sliding angle and thermal conductivity, the filler ratio of nanocomposites was optimized. The test results show that when the content of APT-SiC particles is 20 wt %, super-hydrophobic polyurethane coating can be obtained. The coating has good pull-off strength and thermal conductivity, and the contact angle and sliding angle are 161° and 3°, respectively. In addition, the practical application of the super-hydrophobic polyurethane coating was tested by related experiments. The experimental results show that the coating has good self-cleaning, wear resistance and anti-corrosion performance, can meet the requirements of air coolers in special environments, and has great application prospects.

**Keywords:** super-hydrophobic; waste cooking oil; environmentally sustainable; modified polyurethane; hydrophobic nano-silicon carbide particles; high mechanical durability

## 1. Introduction

With the increase of mining depth, there is an increasing number of high-temperature mines. In China, about 150 coal mines have different degrees of high-temperature heat damage [1]. Usually, ventilation is not enough to solve the problem of mine heat hazard, so it is necessary to add refrigeration system to cool them [2,3]. As the terminal of the refrigeration system, the mine air cooler is generally arranged near the working face. These locations are narrow in space, air pollution is serious, and pollutants such as coal dust are easily attached to the fin surface of the mine air cooler. The presence of pollutants will increase the air side wind resistance and reduces the thermal conductivity of fins, which seriously affects the heat transfer efficiency of the air cooler [4]. In order to make the mine air cooler run efficiently, pollutants must be removed regularly. Existing mine air coolers generally require special cleaning equipment to remove pollutants, which increases additional equipment investment and is damages fins very easily. As far as the present situation is concerned, it is urgent to take measures to reduce the adhesion of pollutants on the fin surface of mine air coolers.

Super-hydrophobic surfaces have attracted extensive research in academia and industry due to their applications in self-cleaning, anti-icing, anti-corrosion and oil-water separation [5–8]. Recent studies by Yang et al. [9] have found that the super-hydrophobic surface has excellent dust removal performance, can significantly reduce the amount of dust attached to the surface of the finned air cooler. There are two essential requirements for the preparation of super-hydrophobic surfaces: one is low surface-energy materials, and the other is micro-nano layered structure [10]. In general, the adhesion between the low surface-energy component and the substrate is relatively poor, and the micro-nanostructure is very fragile [11]. It is easy to damage or even lose the super-hydrophobic properties when the super-hydrophobic surface is affected by mechanical effects such as shock friction. Super-hydrophobic surfaces without high mechanical stability function with difficulty in practical applications [12–14].

Recent studies have shown that the use of a coating-forming polymer such as polyurethane in combination with nanoparticles can improve the durability of super-hydrophobic coatings. Elahe Yousefi et al. [15] used the sol-gel process to prepare a super-hydrophobic and highly oleophobic polyurethane coating. The coating has good tensile hardness, and the contact angle is up to 159°. Tang et al. [16] prepared a super-hydrophobic coating by spraying polyurethane and molybdenum disulfide suspension on various substrates. The coating exhibits good wear resistance, the contact angle exceeds 153°. Cao et al. [17] mixed modified waterborne polyurethane (SiWPU) with hydrophobic silica nanoparticles and obtained a super-hydrophobic coating by one-step spraying. The coating has good mechanical durability and thermal stability, and the heat resistance temperature can reach 100 °C. However, to our knowledge, the polyurethanes used in existing super-hydrophobic coatings are mostly derived from petroleum-based polyols, while petroleum-based polyols are not renewable.

With the depletion of fossil resources and the further understanding of non-biodegradable products, the use of bio-based materials to synthesize waterborne polyurethane (WPU) has received more and more attention [18]. In recent years, many reports have been published on the synthesis of waterborne polyurethane dispersions from vegetable oils, such as linseed oil [19], soybean oil [20], castor oil [21], jatropha oil [22], cottonseed oil [23], Malaya oil [24], sunflower seed oil [25]. Waste cooking oil (WCO) has similar composition to vegetable oil, while the price of waste cooking oil is much lower. In China's small cities (population < 200 k), waste cooking oil is often discharged directly into the environment due to lack of processing facilities, which seriously pollutes the environment. At the same time, some illegal cooking oil producers can recycle waste cooking oil and sell it to restaurants, which poses a serious threat to human health [26,27]. Reasonable use of waste cooking oil can not only reduce production costs, but also avoid the abuse of waste cooking oil, which improves the environment and reduces health problems.

In this context, we propose a super-hydrophobic polyurethane coating for fins of mine air coolers. Considering environmental protection and production cost, polyurethane prepolymer was synthesized from waste cooking oil. Amino-terminated polydimethylsiloxane was used to modify polyurethane prepolymer, which improved the crosslinking ability of polyurethane at room temperature. To address the poor thermal conductivity and poor mechanical properties of polymer coatings, a series of nanocomposites were prepared by the combination of hydrophobic nano-silicon carbide particles and super-hydrophobic polyurethane (SPU) emulsion. The filler ratio of the nanocomposites was optimized by related tests (pull-off strength test, wettability test and thermal conductivity test). In order to ensure that the super-hydrophobic polyurethane coating can be used effectively in harsh conditions for a long time, its practical application performance was evaluated by a self-cleaning performance test, sandpaper wear test and salt solution immersion test.

## 2. Experiment

### 2.1. Materials

Purified waste cooking oil (WCO) was purchased from Shanghai Luming Environmental Protection Technology Co., Ltd. (Shanghai, China). Diethanolamine (DEA) was purchased from Shanghai Shengyu Chemical Co., Ltd. (Shanghai, China). Isophorone diisocyanate (IPDI) and dibutyltin dilaurate (DBTDL) were purchased from Wanhua Chemical Group Co., Ltd. (Shandong, China). Sodium methoxide and butyl acetate was purchased from Shanghai Aladdin Biochemical Technology Co., Ltd. (Shanghai, China). 3-aminopropyl triethoxysilane (APTES) and amino-terminated polydimethylsiloxane (ATP) were purchased from Nanjing Xuanhao New Materials Co., Ltd. (Jiangsu, China). The spherical silicon carbide particles were supplied by Huzhou Yuanqin New Materials Co., Ltd. (Zhejiang, China). Sodium chloride, diethyl ether and ethanol were purchased from Chengdu Cologne Chemical Co., Ltd. (Sichuan, China).

### 2.2. Preparation of Super-Hydrophobic Polyurethane (SPU) Emulsion

Waste cooking oil fatty amide (WCOFA) was synthesized from purified waste cooking oil by amino decomposition reaction [28,29]. First, we placed sodium methanol (0.38 g) and DEA (33.6 g) in a four-neck round-bottom flask with a capacity of 500 mL, then turned on the heating switch and mechanical mixing device and the mixture was stirred at 80 °C for 20 min. WCO (90 g) was added drop by drop within 60 min. After the WCO was completely added, the reaction temperature was raised to 120 °C and the stirring reaction continued for 3 h. The final product was dissolved in diethyl ether and washed with a 15 wt % aqueous solution of sodium chloride. The ether containing WCOFA and sodium chloride aqueous solution were separated by a separating funnel. WCOFA was obtained after decompression was steamed out of ether.

Preparation of SPU emulsion using WCOFA as raw material. First, WCOFA (32.5 g) was mixed with butyl acetate to obtain a 50 wt % WCOFA solution. Then the WCOFA solution, IPDI (26.7 g) and DBTDL (2.65 g) were placed in a four-necked flask. The heating switch and mechanical stirring device were turned on and the mixture was stirred at 80 °C for 1 h to obtain a polyurethane prepolymer. Finally, the reaction temperature was reduced to 55 °C, ATP (10 g) was added into polyurethane prepolymer drop by drop, and the SPU emulsion was obtained by stirring for 3 h. The basic chemical reaction formula was shown in Figure 1.

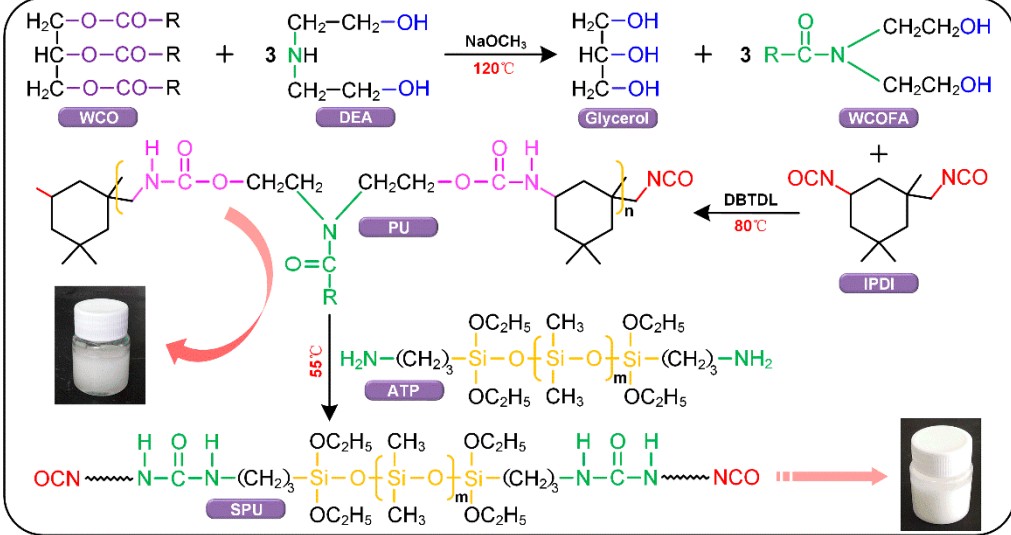

**Figure 1.** Basic chemical reaction formula for synthesizing super-hydrophobic polyurethane (SPU).

### 2.3. Characterizations of SPU and Its Intermediates

#### 2.3.1. Chemical Structure Characterization

The chemical structure of the SPU and its intermediates were inferred based on the position and shape of the spectral absorption peaks. The liquid sample to be tested was coated on a dry KBr wafer, the Fourier transform infrared (FT-IR) spectra of the intermediate and SPU were recorded by Nicolet iS50 spectrometer (Thermo Scientific, Waltham, MA, USA). Each sample was carried out in the range of 4000–500 cm$^{-1}$ with a resolution of 4 cm$^{-1}$. Using tritium chloroform (CDCl$_3$) as solvent and tetramethyl silane (TMS) as internal standard, the $^1$H nuclear magnetic resonance (NMR) spectra of WCOFA were recorded by a 400 MHz AV-300 nuclear magnetic resonance spectrometer (Bruker, Bremen, Germany).

#### 2.3.2. Thermal Stability Test

The thermal stability of cured polymer coatings was measured by TGA 2 thermogravimetric analyzer (Mettler Toledo, Switzerland). The sample of about 10 mg was placed in an alumina disk and increased at the heating rate of 10 °C/min in N$_2$ atmosphere, the terminal reaction temperature was 600 °C. During the pyrolysis process, the thermogravimetric analyzer automatically records the change of the mass with time and temperature, and then performs data processing to obtain the corresponding thermogravimetric-derivative thermogravimetric (TG-DTG) curve.

### 2.4. Preparation of Nanocomposite Coating

In view of the poor mechanical durability of super-hydrophobic surfaces, some super-hydrophobic coatings with nanoparticles and polymers have been proposed in recent years [30]. Nanoparticles such as SiO$_2$ [31], ZnO [32], TiO$_2$ [30] and Al$_2$O$_3$ [33] are often used to manufacture super-hydrophobic coatings. Achieving efficient dispersion of the filler in the polymer matrix is a very important task because filler particles, especially nano-sized filler particles, tend to aggregate due to their high surface energy [34]. Relevant studies show that the agglomeration of SiC in aqueous solution can be controlled effectively by chemical modification of organic or inorganic molecules. [35].

#### 2.4.1. Synthesis of Nanocomposites

In this study, nano-silicon carbide particles were treated by APTES. First, 20 g nano-silicon carbide particles were dispersed in 200 mL ethanol solution by ultrasonic treatment, then 15 mL APTES was added, and modified silicon carbide (APT-SiC) solution was obtained by magnetic stirring at 70 °C for 4 h. The solid product is obtained by removing the solvent through the vacuum pump. The solid product was repeatedly washed with ethanol and purified water to remove excess APTES, and the APT-SiC particles were obtained by drying and grinding [36]. Finally, the obtained APT-SiC particles were added to the SPU emulsion, and after ultrasonic treatment at 40 °C for 30 min, the nanocomposite was obtained. In order to study the effect of the filler ratio on the properties of the nanocomposite, the content of the APT-SiC particles was controlled between 0 and 30 wt %.

#### 2.4.2. Coating Application

In order to ensure a good interaction between the coating and the test piece, the surface of the test piece was pretreated by plasma treatment to remove contaminants and reduce surface roughness before applying nanocomposites. Then, the prepared nanocomposites were coated on the surface of the pretreated test pieces. Finally, the test pieces coated with nanocomposites were placed in a vacuum drying oven, dried and cured at 130 °C for 6 h to obtain an nanocomposite coating. The flow chart of the preparation process of the nanocomposite coating is shown in Figure 2.

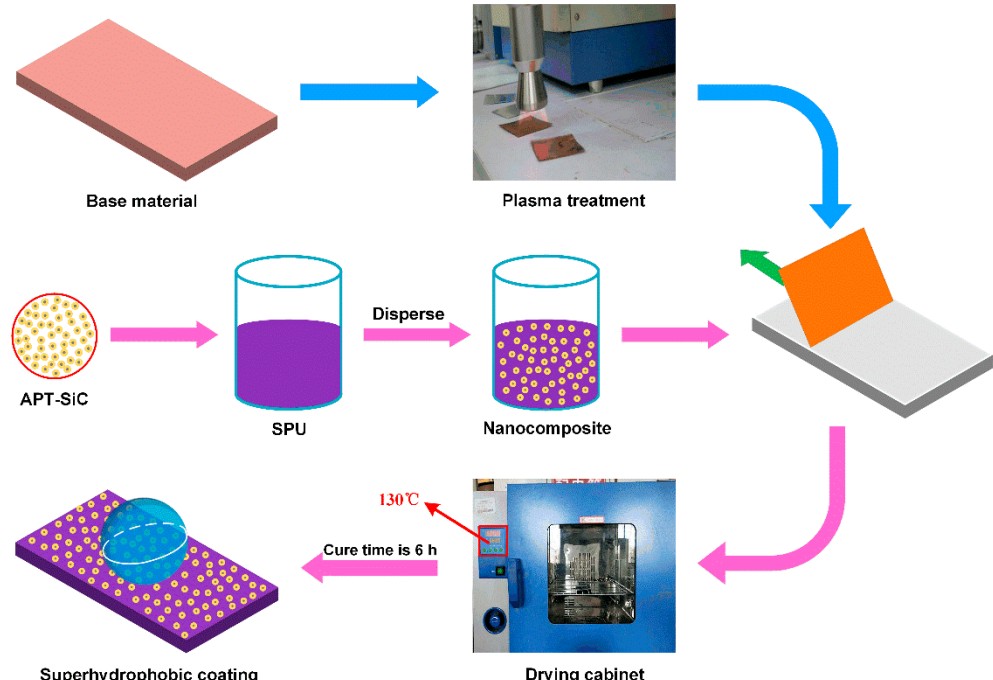

**Figure 2.** Flow chart of the preparation process of the nanocomposite coating.

## 2.5. Characterization of APT-SiC

### 2.5.1. Characterization of Chemical Grafting

For characterization of chemical grafting of APTES on to the APT-SiC surface by FT-IR spectroscopy, firstly, the dried APT-SiC sample was mixed with KBr in the ratio of 1:150, then the mixture was ground into powder, and the transparent sheet was prepared by pressing. FT-IR spectra of APT-SiC were recorded by Nicolet IS50 (Thermo Scientific, Waltham, MA, USA). Each sample was processed in the range of 4000–500 cm$^{-1}$ with a resolution of 4 cm$^{-1}$.

### 2.5.2. Quantitative Characterization of Grafting

The amount of APTES grafted on the surface of APT-SiC was measured by TGA 2 thermogravimetric Analyzer Mettler Toledo and Switzerland. The sample of about 10 mg was placed in the alumina tray and heated at 10 °C/min in N$_2$ atmosphere, the terminal reaction temperature was 800 °C. During pyrolysis, the thermogravimetric analyzer automatically records the mass change with time and temperature, and then performs data processing to obtain the corresponding TG-DTG curve.

### 2.5.3. Characterization of Dispersion Stability

The effective dispersion of APT-SiC is very important to obtain stable super-hydrophobic polyurethane coating. In order to explain the dispersion stability of APT-SiC intuitively, the natural settlement test was used to characterize the APT-SiC. SiC and APT-SiC powders were first dispersed in pure water by ultrasonic treatment and then placed on a horizontal table top. We observed and took macro photos at regular intervals.

## 2.6. Optimization of Filler Ratio of Nanocomposites

Polymer nanocomposites are widely used in various fields due to their excellent mechanical, thermal, electrical, optical and magnetic properties [37]. The related studies show that the improvement of the properties of polymer nanocomposites depends to a large extent on the dispersion of nano-fillers in polymer matrix, the compatibility of fillers and polymers and the interaction between fillers and

polymers [38]. It can be seen that nano-fillers play a crucial role in improving the performance of polymer nanocomposites. In order to obtain the best performing coating, the filler ratio of nanocomposites was optimized by the following experiments.

### 2.6.1. Pull-Off Strength Test

Pull-off strength is an important index to evaluate the properties of the coating. only when the interface between the coating and the substrate is reliable and robust, the coating can better achieve other functions [39]. The pull-off strength of the nanocomposite coating was tested by the PosiTest AT-A digital display adhesion tester (DeFelsko, Ogdensburg, NY, USA). According to the ASTM D4541 [40], the coating surface was first bonded with the spindle by adhesive, then the adhesive was cured at room temperature for 24 h. The cutter was used to cut the coating to the substrate, and finally the spindle was connected to the adhesion tester for testing.

### 2.6.2. Wetting Property Test

In general, the surface with water contact angle (WCA) > 150° and water sliding angle (WSA) ≤ 10° refers to a super-hydrophobic surface [41]. WCA and WSA are important indicators for evaluating the wetting properties of super-hydrophobic surfaces. In this study, the WCA and WSA of nanocomposite coating were measured by a DSA30 contact angle measuring instrument (KRUSS, Germany). The deionized water used to test the WCA were 3 μL, and the WCA values of each sample are determined by the average measured values of five different positions of the coating.

### 2.6.3. Study of Surface Morphology

It is well known that the wetting properties of coatings are closely related to their surface morphology. In order to clearly understand the effect of filler content on the wetting behavior of the coating, the surface morphology of the nanocomposite coating was observed by an Apreo scanning electron microscope (FEI, Hillsboro, OR, USA). Firstly, the prepared nanocomposite coatings were cut into suitable size test samples. Then, the test samples were fixed on the sample table with conductive tape, and a layer of gold was deposited on the surface of the test samples by sputtering. Finally, the test samples were put into the sample room for imaging and observation.

### 2.6.4. Thermal Conductivity Test

Polymers are widely used in industry and daily life because of their versatile functions, light weight, low cost and excellent chemical stability. However, in some applications such as heat exchangers and electronic packaging, the low thermal conductivity of polymers is one of the main technical obstacles [42]. The thermal conductivity of polymer materials is becoming increasingly important to improve the reliability of the device. In order to obtain the best thermal conductivity of the coating, the thermal conductivity of nanocomposite coating s was measured by the thermal conductivity measuring instrument DRL-II (Hunan, China).

### 2.7. Application Performance Test of the Super-Hydrophobic Polyurethane Coating

So far, many methods for creating super-hydrophobic coatings have been reported, but few products have been applied to them in practice [43]. The main reason for this limited application is not the cost or difficulty of mass production, but its poor stability [44]. In order to ensure that the super-hydrophobic polyurethane coating can meet the needs of practical work, the practical application performance of the super-hydrophobic polyurethane coating was tested through the following experiments.

### 2.7.1. Self-Cleaning Performance Test

The self-cleaning performance of a super-hydrophobic surface is directly related to the decontamination effect in the application process. In order to verify the decontamination effect of the super-hydrophobic polyurethane coating in use, the carbon black powder taken from the 1412 coal face of Tangkou Coal Mine was used as a pollutant for a self-cleaning performance test. The sample to be tested was tilted in a petri dish with a small inclination angle, and the sparse carbon black powder layer was evenly sprayed on the surface of the sample. Then, the injector was used to clean the sample through the directional movement of water droplets (see Video S1). A digital camera was used to take pictures to obtain digital images at different times.

### 2.7.2. Wear-Resistance Test

It is unavoidable that the super-hydrophobic coating will contact and experience friction with other objects during use. The mechanical wear resistance is very important for its long-term and effective use [45]. In this study, the wear resistance of the super-hydrophobic polyurethane coating was evaluated by sandpaper wear test. Firstly, the sample was placed on 320 mesh sandpaper to make the coating contact with sandpaper, and a weight of 300 g was placed on the metal surface. The sample was pushed 10 cm in the same direction, and then rotated 180° and moved 10 cm in the opposite direction. Every 20 cm was recorded as a cycle, for a total of 50 cycles (see Video S2). The contact angle of worn samples were measured by DSA30 contact angle measuring instrument. The specific measurement method was the same as above.

### 2.7.3. Corrosion-Resistance Test

The mineral dust contains $SO_2$ acid gas, which reacts with water vapor to form sulfuric acid vapor. When the heat-transfer surface temperature is lower than the sulfuric acid dew point temperature, the sulfuric acid steam condenses on the surface of the air cooler fins and causes corrosion [46]. Therefore, in order to ensure the normal use of fins of mine air coolers, the super-hydrophobic polyurethane coating must have good corrosion resistance. In this study, the salt solution immersion test was used to test the corrosion resistance of the super-hydrophobic polyurethane coating. The 5 wt % NaCl solution was selected as the salt solution. In order to speed up the experiment, the pH of the salt solution was reduced to about 3 with sulfuric acid. The sample was immersed in sulphate solution for 14 days, observed every 48 h, and macroscopic photos were taken.

## 3. Results and Discussion

### 3.1. Characterization of SPU and Its Intermediates

#### 3.1.1. Fourier Transform Infrared (FT-IR) Spectral Analysis

The FT-IR spectrum of WCOFA is shown in Figure 3a. It can be seen from the Figure that there is a wider absorption band at 3380 $cm^{-1}$, which is attributed to the stretching vibration of –OH. The characteristic absorption peaks at 2978 and 2893 $cm^{-1}$ are the stretching vibration peaks of –CH$_3$ and –CH$_2$, respectively. The absorption peak at 1656 $cm^{-1}$ is caused by the stretching vibration of C=O in amide group. At 1486 and 1385 $cm^{-1}$ is the –CH$_2$ deformation vibration peak and the C–N stretching vibration peak at 1223 $cm^{-1}$. The presence of > N–C=O and –OH groups can prove that the synthesized product is WCOFA.

The FT-IR spectra of PU and SPU are shown in Figure 3b. It can be seen from the figure that there are strong peaks at 3307, 1690, 1550 and 1277 $cm^{-1}$, which are attributed to the stretching vibration of N–H in the carbamate group, the stretching vibration of C=O, the deformation vibration of N–H and the stretching vibration of C–O, respectively. The above four absorption peaks prove the existence of carbamate group. It can be seen from the spectrum of SPU that the absorption peak at 3310 $cm^{-1}$ is enhanced and the absorption peak at 2285 $cm^{-1}$ is weakened, which indicates that the NH$_2$ in

ATP has successfully reacted with NCO in polyurethane. New peaks appeared at 1144 cm$^{-1}$ and 815 cm$^{-1}$, which are attributed to the stretching vibration of Si–O–Si and the stretching vibration of Si–C in Si–CH$_3$ group, respectively. It can be known from these analyses that the SPU has been successfully synthesized.

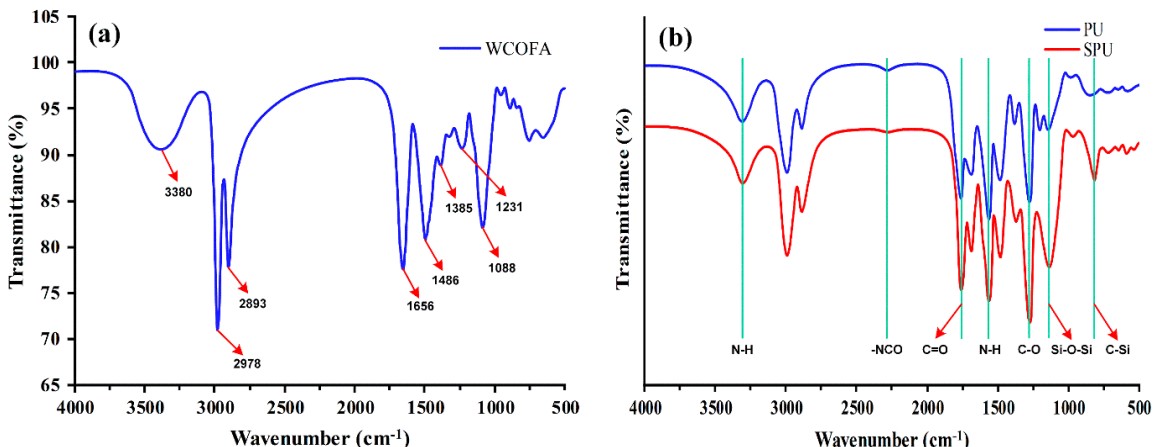

**Figure 3.** Infrared spectrum of the synthesized product: (**a**) waste cooking oil fatty amide (WCOFA), (**b**) PU and SPU.

### 3.1.2. $^1$H Nuclear Magnetic Resonance (NMR) Spectral Analysis

The $^1$HNMR spectrum of WCOFA is shown in Figure 4. As can be seen from the Figure, the peak of CH$_3$ at the end of the fatty acid chain was observed at 0.83 ppm, and the chemical shift of CH$_2$ in the chain appeared at 1.2 ppm. The single peaks at 1.97 ppm and 2.22 ppm correspond to the protons of CH$_2$ linked to olefin and carbonyl, respectively. A single peak of CH$_2$ linked to amide nitrogen was observed at 3.25 ppm. The protons of CH$_2$ near hydroxyl group appeared at 3.5 ppm and the wide single peak of OH was observed at 4.53 ppm. The protons of olefin unsaturation were captured at 5.3 ppm. The proposed WCOFA structure was confirmed by $^1$H NMR spectroscopy.

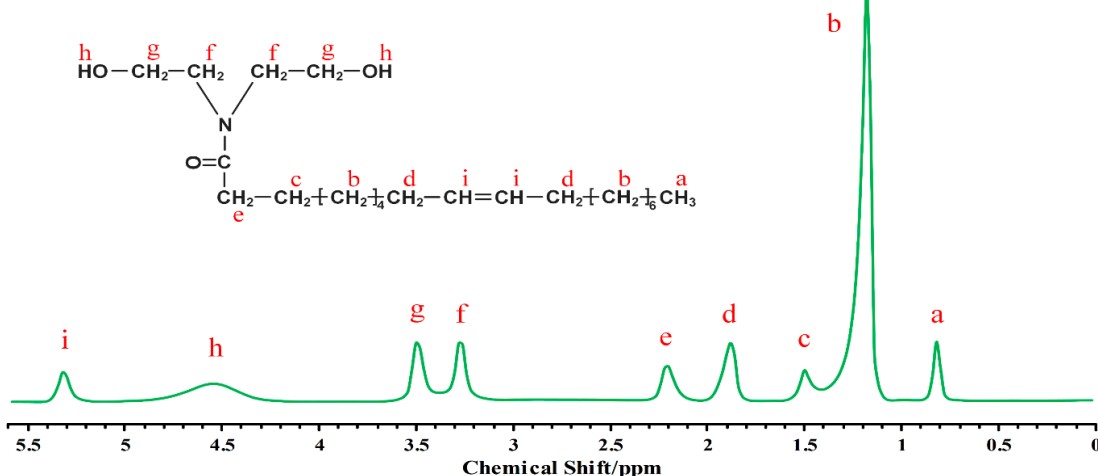

**Figure 4.** $^1$H nuclear magnetic resonance (NMR) spectrum of WCOFA.

### 3.1.3. Thermogravimetric Analysis (TGA)

Figure 5 shows the TG-DTG results of PU and SPU, respectively. From the TG curve, it can be seen that the decomposition of PU is mainly divided into two stages. The first stage of degradation occurs at 200–298 °C, which is mainly attributed to the degradation of carbamate and urea bond in hard segment.

The second stage of degradation occurs at 298–424 °C, which is related to the degradation of soft segment WCOFA. Similar to PU, the decomposition of SPU is mainly divided into two stages: the first stage of degradation occurred at 238–336 °C, the second stage of degradation occurred at 336–470 °C. However, compared to PU, the $T_{10}$ of the SPU increases from 277 °C to 314 °C, and the $T_{50}$ change from 351 °C to 397 °C. This situation is related to the addition of ATP. The addition of ATP firstly forms a crosslinked network structure between PU macromolecular chains, which greatly enhances the intermolecular force, secondly reduces the amount of more combustible organic components and produces siliceous residual substances that inhibit heat and aggregate transfer [19,47]. Through the above analysis, it is known that the SPU has better thermal stability than the PU.

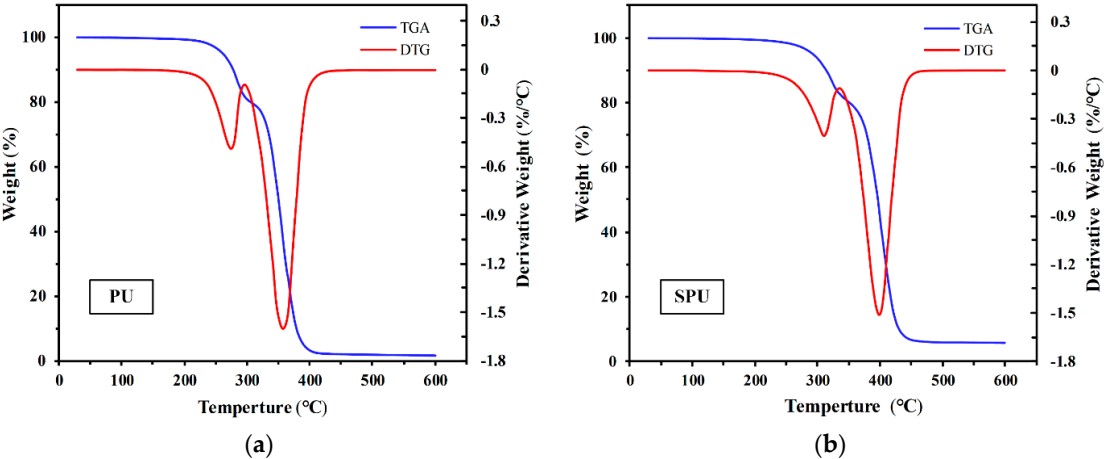

**Figure 5.** Thermogravimetric-derivative thermogravimetric (TG-DTG) curve of PU (**a**) and SPU (**b**).

*3.2. Characterization of APT-SiC*

3.2.1. Chemical Graft Analysis

The FT-IR spectra of SiC and APT-SiC are shown in Figure 6. It can be seen from the figure that SiC exhibits an absorption peak at 825 cm$^{-1}$, which is attributed to the stretching vibration of Si–C. In contrast, APT-SiC showed new absorption peaks at 2958, 2901, and 1113 cm$^{-1}$, which were attributed to the stretching vibrations of −CH$_2$, −CH, and Si–O in APTES. The absorption peak at 825 cm$^{-1}$ is additionally significantly strengthened, which is a result of the superposition of Si–C in APTES and SiC in SiC. From these analyses, APTES is grafted to the SIC surface by chemical bond rather than physical adsorption.

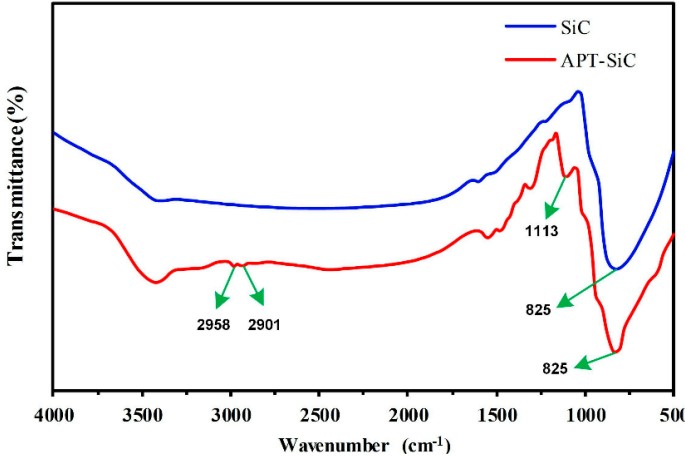

**Figure 6.** Fourier transform infrared (FT-IR) spectra of SiC and APT-SiC.



### 3.2.2. Analysis of Graft Quantity

Figure 7 shows the thermogravimetric curves of SiC and APT-SiC. The thermogravimetry curve of Sic shows that SiC has good thermal stability, and its mass will not change with the increase of temperature. The maximum weight loss ratio is only 0.67%. Compared with SiC, the mass of APT-SiC begins to decrease obviously after the temperature rises to 190 °C, and the maximum weight loss ratio is 6.08%, which is caused by APTES decomposition on the surface of APT-SiC. The results show that APTES has been successfully grafted onto the SiC surface.

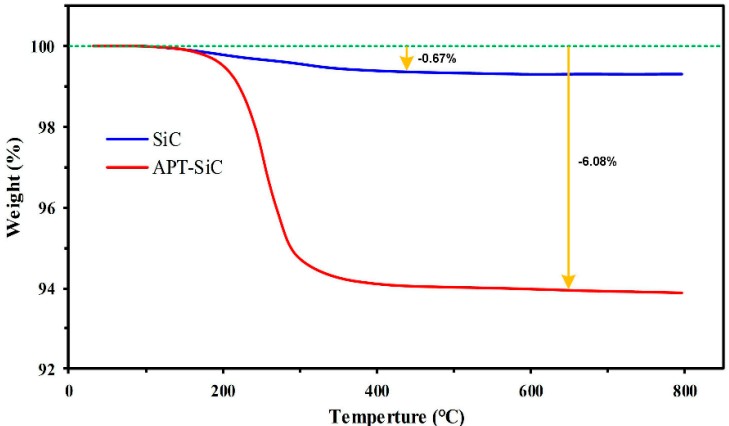

**Figure 7.** Thermogravimetric curves of SiC and APT-SiC.

### 3.2.3. Analysis of Dispersion Stability

Figure 8 shows the sedimentation images of the dispersions at different times. It can be seen from Figure 8 that both dispersions have good dispersion at the initial stage. The SiC dispersion was suspended in gray, while the APT-SiC dispersion was suspended in yellow-gray, which was attributed to the surface modification of SiC by APTES. The SiC dispersion showed slight precipitation after 1 h as the standing time progressed. After standing for 5 h, the precipitation at the bottom of the SiC dispersion further increased and the upper part showed a translucent state. After standing for 12 h, the SiC dispersion had completely precipitated. In contrast, the APT-SiC dispersions were consistently able to maintain good dispersion, and no significant precipitation was observed at the bottom. These results confirm the good dispersion stability of ATP-SiC.

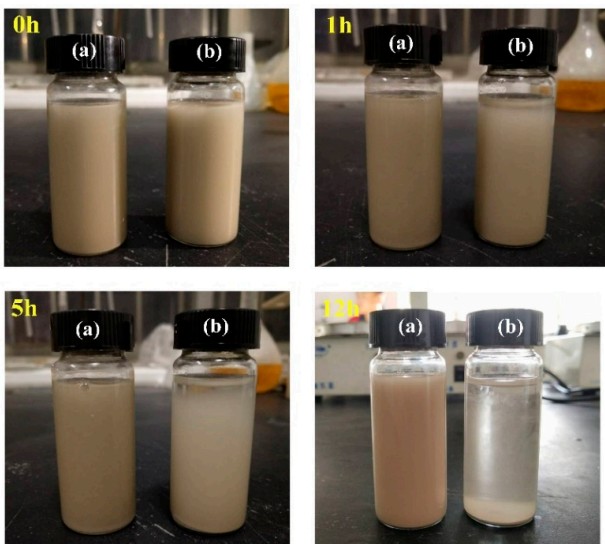

**Figure 8.** Dispersion stability of APT-SiC (**a**) and SiC (**b**) in water at different times.

### 3.3. Determination of the Optimal Ratio of Nanocomposites

#### 3.3.1. Pull-Off Strength Test

Figure 9a,b show the adhesion tester and the pull-off strength test results of the nanocomposite coating, respectively. It can be seen from Figure 9b that the pull-off strength of the SPU coating modified by the nano-filler has been improved. At the same filler content, the APT-SiC modified SPU coating has a higher pull-off strength. This is attributed to the good dispersibility of APT-SiC and the strong interaction between the surface amino group and the SPU matrix, which enhances the network structure of the coating and improves the bonding ability between the coating and the substrate. With the increase of APT-SiC content, the pull-off strength of nanocomposite coatings increases continuously. When the content of APT-SiC is 20 wt %, the maximum value is 8.5 MPa. However, when the content of APT-SiC is further increased, the pull-off strength of the nanocomposite coating is decreased. This situation can be explained as follows: as the content of APT-SiC increases, the content of SPU matrix decreases in relative terms, and the adhesion between APT-SiC particles decreases, which promotes the development of cracks and the shedding of coating.

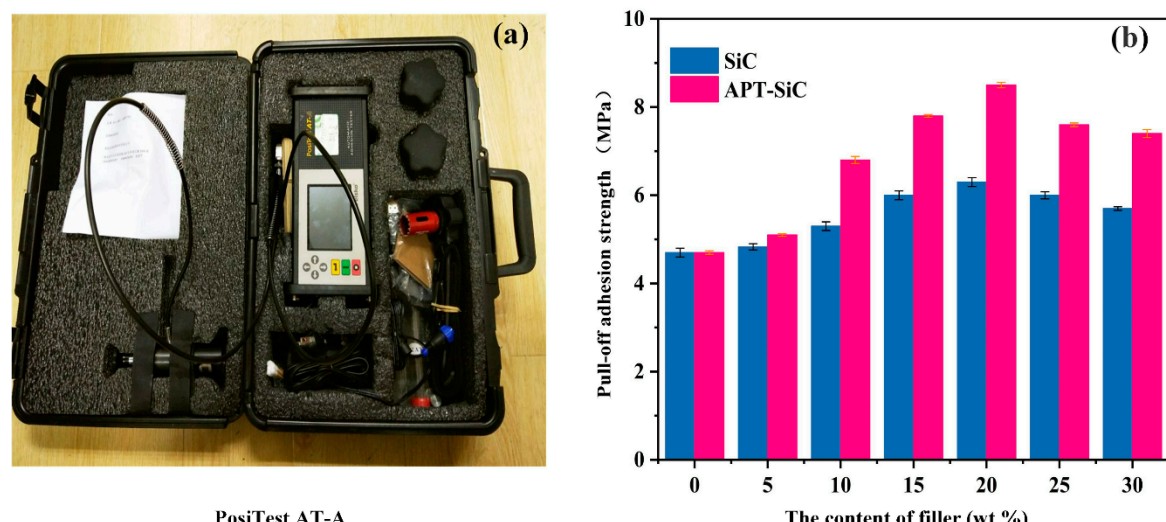

**Figure 9.** (**a**) Adhesion tester and (**b**) pull-off strength test results of nanocomposite coating.

#### 3.3.2. Study on Wetting Properties and Surface Morphology

The measurement results of the contact angle and sliding angle of the nanocomposite coating are shown in Figure 10. It can be seen from the figure that the SPU coating without nano-filler has good wetting properties, the contact angle and sliding angle are 101° and 28°, respectively. This is because the SPU contains a large amount of silicone chains that can be transferred to the air side and enriched on the cured surface, thereby reducing the surface energy of the coating [17]. At the same filler content, the SPU coating modified by APT-SiC has better wettability. This is attributed to the good dispersibility of APT-SiC, which effectively reduces the aggregation between the nano-particles, thus creating a uniform roughness structure on the surface of the SPU coating.

With the increase of APT-SiC content, the contact angle of nanocomposite coating increases first and then decreases, and the sliding angle decreases first and then increases. When the APT-SiC content is 20 wt %, the wettability of nanocomposite coatings is the best, and the contact angle and sliding angle are 161° and 3°, respectively. This phenomenon can be further explained by the SEM image of the nanocomposite coating. It can be seen from the observation of Figure 11 that the content of APT-SiC has an important influence on the surface morphology of the nanocomposite coating. The surface of the SPU coating without APT-SiC is very smooth. With the increase of APT-SiC content, the nanoparticles gradually rise to the surface of the SPU coating, forming a uniform roughness structure,

thus improving the wetting properties of the coating. When the content of APT-SiC further increases, the surface of the SPU coating is wrapped in dense nanoparticles, and the roughness is reduced to some extent, resulting in a decrease in the wetting performance of the coating.

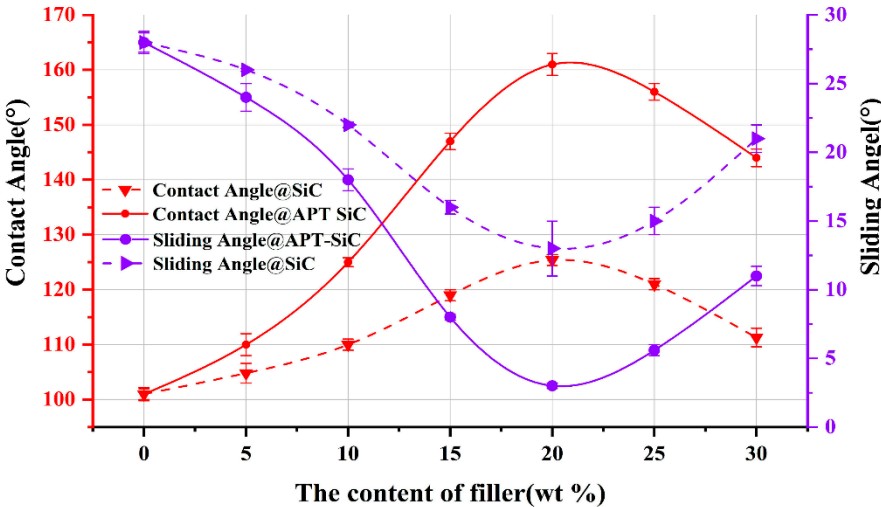

**Figure 10.** Curve of contact angle and sliding angle of nanocomposite coating.

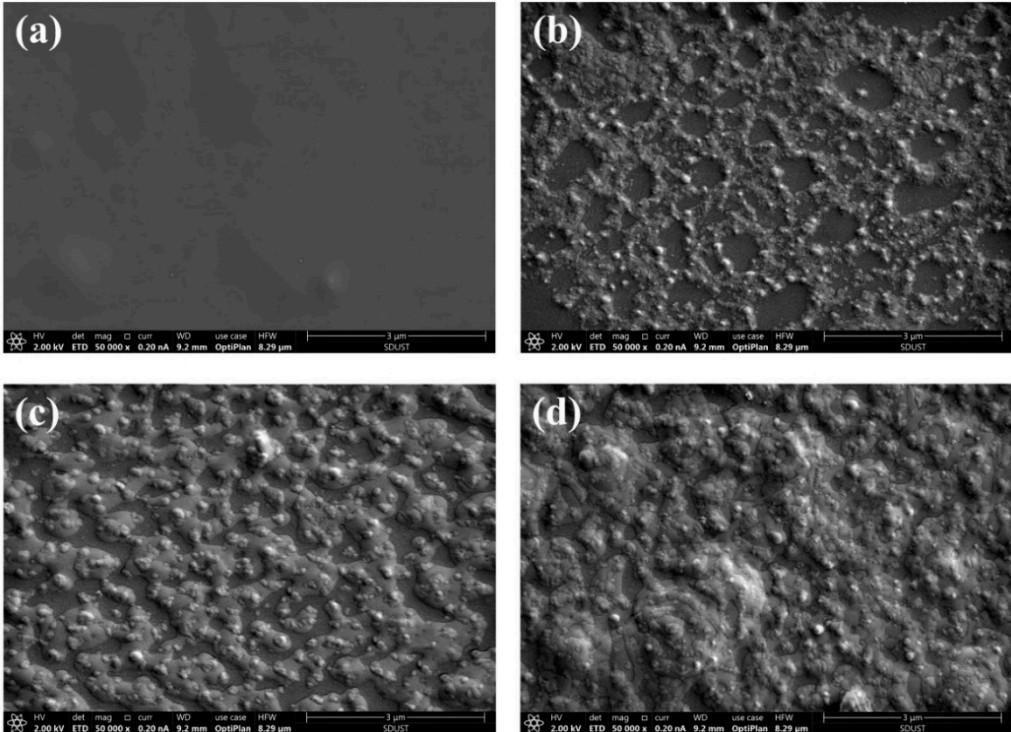

**Figure 11.** Scanning electron microscope (SEM) images of the surface of the investigated samples: APT-SiC@ 0 wt % (**a**), APT-SiC@ 10 wt % (**b**), APT-SiC@ 20 wt % (**c**), APT-SiC@ 30 wt % (**d**).

### 3.3.3. Thermal Conductivity Test

Figure 12 shows the thermal conductivity test results of the nanocomposite coating. As can be seen from Figure 12, the SPU coating modified by APT-SiC particles has higher thermal conductivity than untreated SiC particles. This situation can be explained in two ways. On the one hand, the good dispersibility of APT-SiC particles make more connection paths between the components. On the other hand, the close interaction of APT-SiC particles with SPU reduces the thermal resistance of the

interface between the filler and the matrix. When the APT-SiC content is relatively low, the thermal conductivity of the nanocomposite coating increases slowly. As the APT-SiC content further increases to 10 wt %, the thermal conductivity of the nanocomposite coating increases rapidly. However, when the APT-SiC content is more than 20 wt %, the thermal conductivity of the nanocomposite coating increases slowly again. This phenomenon can be explained as follows: when the content of APT-SiC is low, APT-SiC particles are dispersed independently in the SPU matrix and hardly contact each other. Although the thermal conductivity of the local area of the nanocomposites has been improved, the thermal conductivity of the whole material has hardly increased. With the increase of APT-SiC content, APT-SiC particles begin to contact and form chains and heat conduction networks through which a large amount of heat can be released, and the thermal conductivity of the nanocomposite coatings increases rapidly. When the content of APT-SiC further increases, the thermal conductivity path tends to be saturated, and the thermal conductivity of the nanocomposite coatings increases again slowly. This phenomenon can be explained as follows: when the content of APT-SiC is low, APT-SiC particles are dispersed independently in the SPU matrix and hardly contact each other. Although the thermal conductivity of the local area of nanocomposites has been improved, the thermal conductivity of the whole material has hardly increased. With the increase of APT-SiC content, APT-SiC particles begin to contact and form heat-conduction networks. Through these heat transfer networks, a large amount of heat can be released, and the thermal conductivity of nanocomposite coatings increases rapidly. When the content of APT-SiC further increases, the thermal conductivity path tends to be saturated, and the thermal conductivity of the nanocomposite coating increases slowly again.

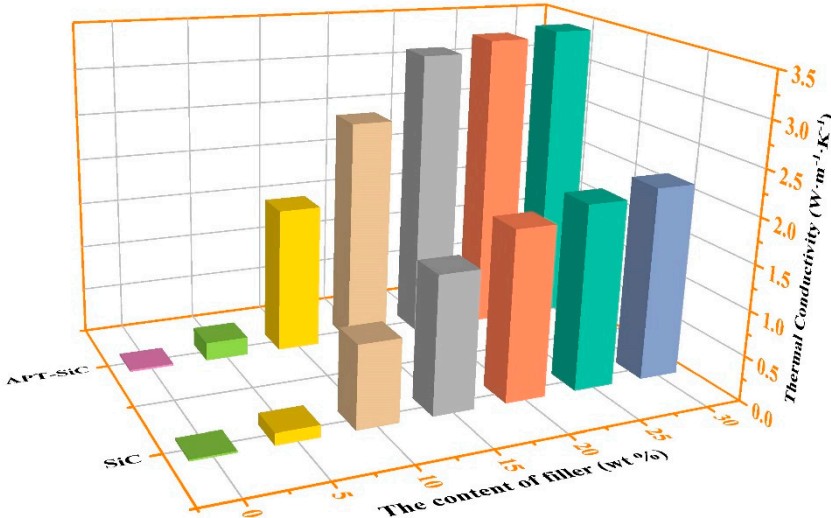

**Figure 12.** Measurement results of thermal conductivity of nanocomposite coating.

From the above test results, it can be seen that the SPU coating modified by APT-SiC particles has better tensile strength, wetting property and thermal conductivity under the same filler content. This indicates that the design structure of the SPU matrix and APT-SiC particles can effectively improve the properties of nanocomposites. When the content of APT-SiC is 20 wt %, the overall performance of nanocomposites is optimal.

*3.4. Study on Application Performance of the Super-Hydrophobic Polyurethane Coating*

3.4.1. Self-Cleaning Performance Test

Figure 13 shows the results of self-cleaning performance test of the polyurethane super-hydrophobic coating. The test sample in the figure consists of two parts, the upper part is the super-hydrophobic polyurethane coating and the lower part is a common substrate surface. As can be seen from the Figure, when the water droplets flow through the polyurethane super-hydrophobic coating covered

with coal powder, the water droplets can roll off in a ball shape and carry away contaminants, leaving a clean surface. In contrast, when water droplets carrying contaminants flow through the surface of a clean common substrate, the water droplets are flat and cannot continue to roll, and the contaminants remain on the surface of the clean substrate. The test results show that the super-hydrophobic polyurethane coating has good self-cleaning performance. The excellent self-cleaning performance of the super-hydrophobic polyurethane coating is attributed to its Lower roll angle, which has been confirmed in previous studies.

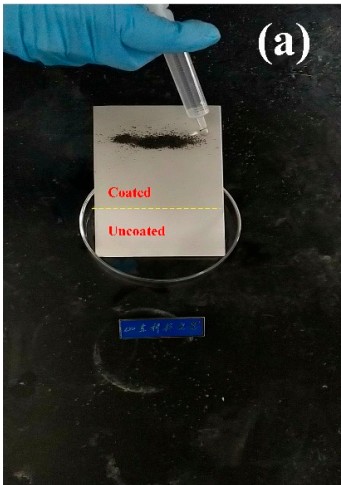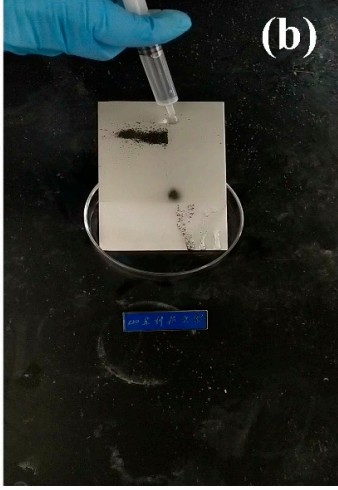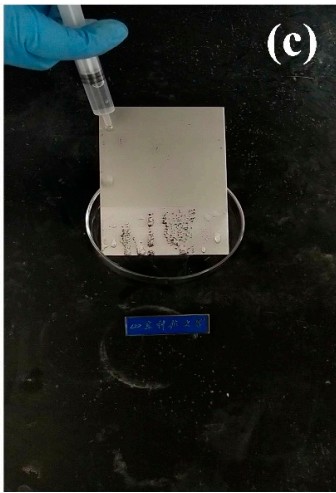

**Figure 13.** Self-cleaning process of the super-hydrophobic polyurethane coating. (**a**) Before the test; (**b**) During the test; (**c**) After the test.

### 3.4.2. Wear-Resistance Test

Figure 14 is a schematic diagram of the wear test and the results of the wear resistance test. From the test results of wear resistance, it can be seen that with the increase of friction distance, the contact angle of the super-hydrophobic polyurethane coating decreases gradually, and the contact angle is always more than 150°. Under the load of 300 g, after 1000 cm wear test, the super-hydrophobic polyurethane coating surface only showed slight scratches, and the water droplets on the surface of the coating were still spherical and do not wet the surface. These results are sufficient to demonstrate that the super-hydrophobic polyurethane coating has good mechanical properties. The excellent wear resistance of the super-hydrophobic polyurethane coatings can be attributed to the following two aspects. On one hand, the ethoxylated hybrid molecules in the SPU can be converted into a hydrophilic silanol group after hydrolysis in the presence of water, and the silanol groups can form covalent bonds through the interaction between hydrogen bonds and metal hydroxide groups existing on metal substrates, which improves the binding ability between coating and substrates [48]. On the other hand, APT-SiC particles have good dispersibility, and the amino group on the surface can form strong chemical bond with SPU chains, which enhances the cohesive strength of the coating.

### 3.4.3. Anti-Corrosive Performance Test

The corrosion resistance test results of different coatings are shown in Figure 15. As can be seen from the Figure, after 6 days of immersion, bubbles and corrosion spots appeared on the surface of the PU coating. This indicates that the salt solution has penetrated into the polyurethane coating and corroded the metal substrate. After 14 days of immersion, bubbles and corrosion spots have already covered the surface of the PU coating. For the SPU coating, after 14 days of immersion, only a few corrosion spots appear on the surface without bubbles. This is because the structure of Si–O–Si network in the SPU chain has a good crosslinking density, which can form a dense coating on the metal surface, thus slowing the diffusion of the corrosive medium to the coating/metal interface [44].

In contrast, there are no bubbles or corrosion spots on the surface of the super-hydrophobic polyurethane coating. These observations demonstrate that the super-hydrophobic polyurethane coating can protect the metal substrate from corrosion in salt solution. The excellent anti-corrosion properties of the super-hydrophobic polyurethane coatings is closely related to the addition of APT-SiC particles. On the one hand, the addition of APT-SiC particles reduces the porosity of SPU coating, making it difficult for electrolytes to penetrate into the metal substrate. On the other hand, the surface roughness of the SPU coating was improved by adding APT-SiC particles. This rough surface structure can capture the air in the groove, reduce the contact area at the solid–liquid interface and thus be more waterproof [49].

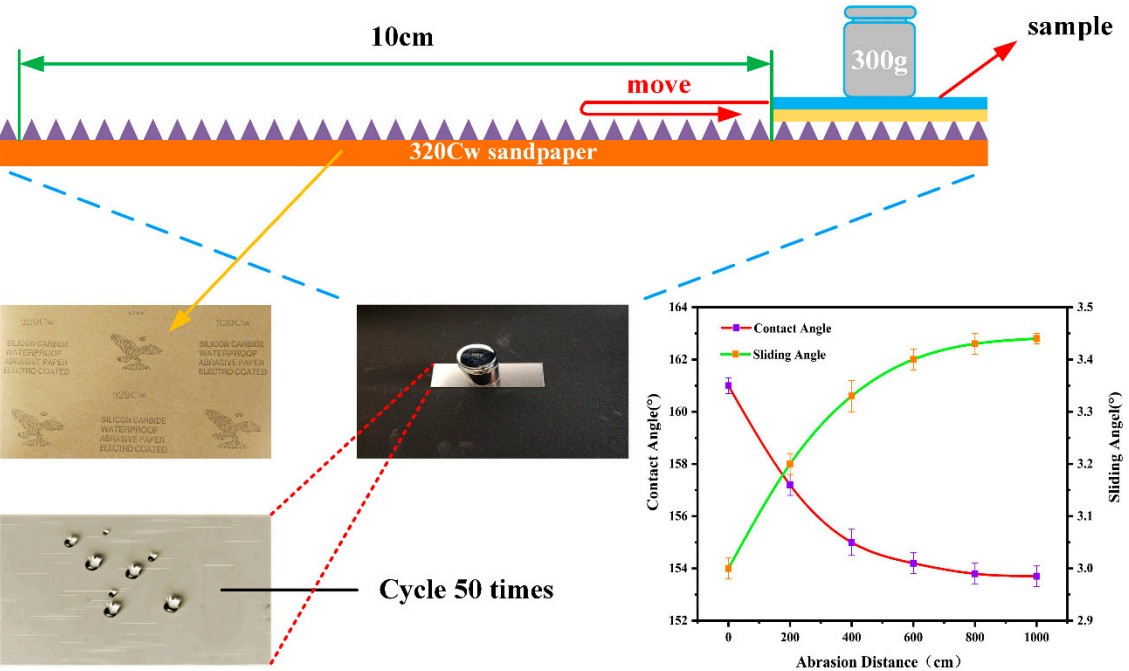

**Figure 14.** Schematic diagram of wear test and test results of wear resistance.

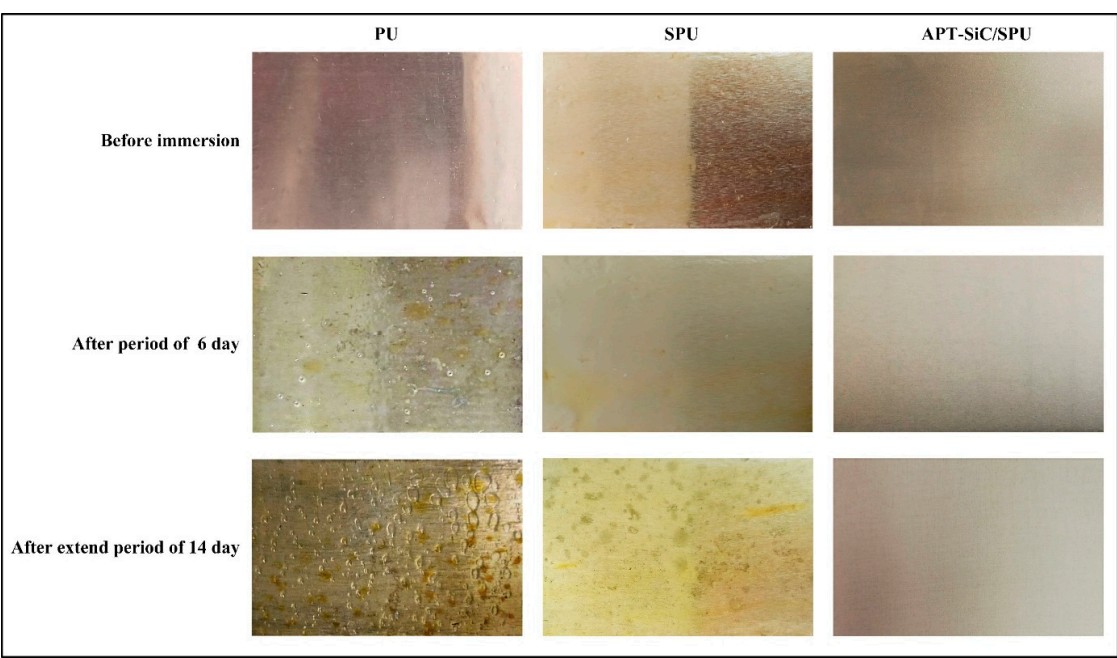

**Figure 15.** Surface topography of the coating after different corrosion times.

## 4. Conclusions

In this work, we successfully synthesized polyurethane prepolymers from waste cooking oil as a raw material, turning waste into treasure and providing a new idea for bio-based materials to synthesize polyurethane. The polyurethane prepolymer was modified by ATP to obtain SPU emulsion, and its chemical structure was successfully characterized by FT-IR spectroscopy. Thermogravimetric analysis showed that SPU has good thermal stability and can remain stable at 220 °C. The designed structure of APT-SiC particles and the SPU matrix effectively improves the overall performance of nanocomposite coatings. when the content of APT-SiC particles is 20 wt %, super-hydrophobic polyurethane coating can be obtained. The coating has good pull-off strength and thermal conductivity, and the contact angle and sliding angle are 161° and 3°, respectively. In addition, the super-hydrophobic polyurethane coating has good self-cleaning, wear resistance and anti-corrosion performance in the application performance test, which can meet the working requirements of the air cooler in a special environment, and has great application prospects.

**Supplementary Materials:** The following are available online at http://www.mdpi.com/2079-6412/9/12/861/s1: Video S1: Self-cleaning performance test process, Video S2: Wear-Resistance test process.

**Author Contributions:** "conceptualization, L.K. and J.J.; methodology, Y.C. and L.K.; writing—original draft preparation, Z.G.; writing—review and editing, Y.C., D.M. and J.J.; supervision, Y.C. and D.M.; project administration, D.M. and Z.G.; funding acquisition, D.M.

**Funding:** The research was funded by the Shandong Provincial Department of Science and Technology under grant number 2018GGX04008.

**Conflicts of Interest:** The authors declare no conflict of interest.

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
