# Peer review of "Preparation and Performance Test of the Super-Hydrophobic Polyurethane Coating Based on Waste Cooking Oil"

_coatings, doi:10.3390/coatings9120861_

Round 1

Reviewer 1 Report

Preparation and performance test of the super- hydrophobic polyurethane coating based on waste cooking oil.

The research carried out in this manuscript is very interesting and can contribute to solve a critical problem in some mining environments. A good introduction is followed by an excellent detailed description of the experimental procedure. Results and discussion are clearly exposed. The weaker parts are 3.2 Determination of the optimum ratio of nanocomposites and 3.3 Study on application performance of the superhydrophobic polyurethane coating. The conclusions are coherent with the results and discussion.

Detailed review:

In my opinion Figure 1 is too “packed”.

Figure 3 is unnecessary.

In paragraph 2.6 I don’t know the interest to mention ultrahydrophobic surfaces.

In 3.1.1 FT-IR characterization: In figure 4 there are some bands non identified: in 4.a below 750 cm-1 ; in 4b around 3000 cm-1 . Are these  from C sp3 ?

In 3.1.2 HNMR characterization: In figure 5: it would be very useful to consider the integrals of peaks b, d and f (mainly) for a real description of protons and hence to identify each group.

Line 292 – 294- 296-297-298-300-409-411: should be APT-SiC instead of ATP-SiC?

Line 298 and figure 7: MPa instead of Mpa

Lines 299-300-301: It would be interesting to show the surfaces after the pull-off test in order to prove the explanation for the lost of adhesion strength.

Line 306 ..It can be seen from the Figureure that … This error can also be found below this line.

Lines 340 to 344 are repeated at lines 348 to 356 with some little changes.

Paragraph 3.3.1: Coal powder used for the self-cleaning behavior, which is the key part of the application, should be analyzed with more detail. For example has the size and morphology of the powder any influence on the efficiency of self cleaning coating? The real environment concerning mining pollutants and conditions of these particles should be taken into account ( flying rate, concentration, size of the solid particles) for a real application of the coating.

Lines 374-376: The low surface energy and the particular roughness of the surface are responsible for the hydrophobic property; the self-cleaning property is mainly related to the low sliding angle shown by the coatings.[B. Bushan, Y.C. Jung. Prog.Mat. Sci. 56 (2010) 1-108]    

In paragraph 3.3.2: Why the load applied was 300g ?  

Reviewer 2 Report

Please see comments and suggestions in the attachement that follows.
